# Isolation, identification, and biological characteristics of *Clostridium sartagoforme* from rabbit

**Ruiguang Gong**[1], **Xiangyang Ye**[2], **Shuhui Wang**[1], **Zhanjun Ren**[1]*

1 College of Animal Science and Technology, Northwest A and F University, Yangling Shaanxi, China,
2 Institute of Animal Science, Guangdong Academy of Agricultural Sciences, State Key Laboratory of Livestock and Poultry Breeding, Key Laboratory of Animal Nutrition and Feed Science in South China, Ministry of Agriculture and Rural Affairs, Guangdong Key Laboratory of Animal Breeding and Nutrition, Guanghzou, China

* renzhanjun@nwafu.edu.cn

**Data Availability Statement:** All relevant data are within the manuscript and its S1 Data, S1 Fig and S1 Raw images files.

**Funding:** This study received funding from the following sources, all awarded to RZ: Shaanxi

## Abstract

In order to develop microbial additives for rabbit feed, a spore-forming bacteria was isolated from the feces of Hyla rabbit using reinforced clostridium medium (RCM). The 16S rDNA sequence of the bacterium was subjected to pairwise sequence alignment using BLAST; the colony morphology, and physiological, biochemical, and stress resistance were studied. The results showed that the bacterium was *Clostridium sartagoforme*, a gram positive anaerobe, which can produce spores. The colony diameter was 0.5 mm—2.5 mm, the diameter of the bacteria was 0.5 μm—1.0 μm × 2.0 μm—6.3 μm, and the spore diameter was 1 μm—1.2 μm × 1 μm—1.2 μm. *C. sartagoforme* can utilize various sugars and alcohols such as fructose, galactose, sorbitol, and inositol. It secreted cellulase into the extracellular environment to form a transparent hydrolysis circle in Congo red medium, it could not liquify gelatin, and the lysine decarboxylase reaction was positive. In liquid medium it entered the stable growth period after 9 h of inoculation. Additionally, it had good stress resistance with a survival rate that exceeded 53% after gastric juice (pH 2.5) treatment for 3 h, it grew in a medium with a bile salt concentration of 0.3%, and the survival rate exceeded 85% after 10 minutes at 80°C. Moreover, animal testing indicated that this strain has no adverse effects on the morbidity and mortality of rabbits. In summary, *C. sartagoforme XN-T4* was isolated from rabbit feces. This bacterium has good resistance to stress, can decompose a variety of monosaccharides and polysaccharides including cellulose, which is relatively harmless for animal health.

## Introduction

Rabbits are herbivores, there are abundant cellulose degrading bacteria resources in rabbit intestinal tract, because they rely on the microorganisms in the caecum to digest cellulose but not produce cellulase by themselves [1]. In addition, cellulose degrading bacteria isolated from animals have natural advantages in the utilization of crude fiber feed. Most of them are

Province Science and Technology Project (K3310216062); Shaanxi Province Agricultural Science and Technology Innovation and Research Project (16NY-108); Shaanxi Province Agricultural Science and Technology Innovation Transformation Project (NYKJ-2020-YL-16); Yangling Demonstration Zone Industry-University-Research Collaborative Innovation Major Project (1017cxy-15); Key R&D project of Shaanxi Province (2018ZDXM-NY-041)].

**Competing interests:** The authors have declared that no competing interests exist.

inherent bacteria in the intestinal tract, which are easier to adapt to the intestinal environment of the animal and play a role [2].

*Clostridium sartagoforme* is an anaerobic spore producing bacterium [3, 4], which determines that the bacterium has a natural advantage in resisting the effects of digestive juice and adapting to the anaerobic environment of the gastrointestinal tract [5]. Researches have shown that the strain can decompose microcrystalline cellulose, carboxymethyl cellulose and chitin [4, 6, 7], and its metabolites include formic acid, acetic acid, propionic acid, lactic acid, butyric acid, isovaleric acid and hydrogen [4, 8, 9]. Simůnek J et al. isolated *C. sartagoforme* from feces of horse, and this strain has high exocellular activity of N-Acetylglucosaminidase [6]. Zhang JN isolated *Clostridium sartagoforme FZ11* from cow dung compost, using urea as a nitrogen source and raw corn stalk as a substrate, confirming that this strain could decompose cellulose and have good hydrogen production characteristics [7]. In the research of Nathani Neelam M and and his coleagues, *Clostridium sartagoforme AAU1* was isolated from Surti buffalo rumen liquor, and they pointed out that VFAs produced by the bacteria play an important role in animal health maintenance through genomic analysis [10]. At present, the research about *C. sartagoforme* isolated from digestive tract or feces of rabbits has not been reported, and there is little information on the biological characteristics of this strain. In this experiment, *C. sartagoforme* was isolated and identified from rabbit feces, and its physiological, biochemical and stress resistance characteristics were preliminarily studied, which help to provide reference for the further development of rabbit microbial additives.

## Materials and methods

### Samples

Fresh fecal samples were taken from Hyla, Rex, and Hycole rabbits at Tongteng Biotechnology Co., Ltd., Yangling Tianxin Rabbit Industry Co., Ltd., and Baoji Enyi Rabbit Farm, respectively. Samples were put into individual sterilized tubes, placed in liquid nitrogen, and rapidly returned to the laboratory.

### Media

Reinforced Clostridium Medium (RCM) contained the following ingredients: peptone (10.00 g), beef extract (10.00 g), glucose (5.00 g), sodium chloride (5.00 g), yeast powder (3.00 g), sodium acetate (3.00 g), soluble starch (1.00 g), L-cysteine hydrochloride (0.50 g), agar (20.00 g) (added in solid medium), and distilled water to bring the final volume of the solution to 1.00 L.

Congo red-sodium carboxymethyl cellulose medium (CMC) was prepared with Congo Red (0.20 g), $KH_2PO_4$ (2.00 g), $MgSO_4$ (0.15 g), agar (15.00 g), yeast powder (2.00 g), sodium carboxymethyl cellulose (10.00 g), and distilled water to bring the solution to a final volume of 1.00 L.

### Isolation and culture methods

A 10 g sample of the collected rabbit feces was placed in a sterilized beaker with 90 mL PBS. The beaker was placed in a 80˚C water bath for 10 minutes to kill the non-spore bacteria; the sample was transferred to 400 mL of RCM liquid medium, sealed with sterilized liquid paraffin, and cultured at 37˚C for 36 h. After culture, the concentrated bacterial solution was diluted to $10^{-3}$, $10^{-5}$, $10^{-7}$. The diluted solutions were applied to the RCM solid medium, the petri dishes were placed into a sterilized container that could be tightly sealed, oxygen was removed by excessive pyrogallic acid, and the dishes were cultured at 37˚C for 48 h. The consumption of pyrogallic acid for deaeration was calculated according to the following chemical reaction

equation:

$$4C_6H_3(OH)_3 + 12NaOH + O_2 = 4C_6H_2(ONa)_3 + 14H_2O$$

After culture, according to the colony morphology of *C. sartagoforme*, colonies were selected for microscopic examination; the colonies that met the characteristics continued to be anaerobic cultured with RCM solid medium (37˚C, 48 h). This step was repeated until a mono-clonal colony was obtained.

## Identification of *C. Sartagoforme*

**Morphological identification.**   Identification was performed according to the colony mor-phology and Gram staining characteristics of the monoclonal colonies. Additionally, the culture time of the medium used for Gram staining was controlled at around 24 h. *Clostridium* is gener-ally positive for Gram staining, but the cell wall structure of the old bacteria would become loose if the culture time is too long, and the result of Gram negative may appear after staining [11].

**16S rDNA gene sequence analysis and identification of target strain.**   The target strains' DNA was extracted by a bacteria genomic DNA extraction kit (Takara Bio Inc., Shiga, Japan). The quality of genomic DNA was determined by 1% agarose gel electrophoresis. The concen-tration and purity of DNA were determined by ultra-micro nucleic acid analyzer. PCR amplifi-cation was carried out according to the required DNA samples. The following 16S r DNA PCR universal primers were used for amplification: 7F- CAGAGTTTGATCCTGGCT and 1540R-AGGAGGTGATCCAGCCGCA. The PCR conditions are shown in Table 1.

PCR fragments were synthesized at the following time-temperature regimes: initial dena-turation at 94˚C for 5 min; subsequent 34 cycles at 94˚C for 30 s, at 58˚C for 30 s, and at 72˚C for 30 s; final polymerization at 72˚C for 2 min.

The PCR products were detected by 1% agarose gel electrophoresis, and the products with a fragment of about 1,500 bp were sent to Beijing AUGCT Company for sequencing. The sequencing results were run through NCBI for BLAST homology comparison and a phyloge-netic tree was constructed; the comparison results were combined with the phylogenetic tree to determine the strain species.

## Physiological and biochemical characteristics of *C. sartargoforme*

**Gelatin liquefaction and sugar fermentation characteristics.**   The physiological and bio-chemical characteristics of the target strain were studied by micro-biochemical reaction identi-fication tube for bacteria (HOPEBIO, Qingdao, Shandong). The bacteria were inoculated in the clean bench, and the biochemical reaction tube was liquid sealed with sterilized liquid par-affin and covered with parafilm.

**Table 1.  *C. sartagoforme* 16S rDNA PCR reaction system.**

| Materials | Volume |
|---|---|
| 2 × Mix | 7.5 μL |
| Template DNA | 0.5 μL |
| Forward Primer | 0.5 μL |
| Reverse Primer | 0.5 μL |
| DD H$_2$O | 6 μL |
| Total Volume | 15 μL |

Note: 2 × Mix (Purchased from ComWin Biotech Co.,Ltd., Beijing).

**Fiber decomposition characteristics.** The target strain was inoculated on CMC-Congo Red plate for anaerobic culture, and the fiber degradation ability of the strain was determined by the fiber hydrolysis circle on the medium.

**Growth and pH curves.** In order to generate the growth and pH curves, 96 tubes containing 10 mL RCM liquid medium, 72 of which were inoculated with target strain at 1% of inoculation volume, were cultured in anaerobic conditions at 37˚C. Three tubes were taken out every 1 h during this period. Three tubes with inoculated bacteria and one uninoculated tube (blank reference) were paired to measure the $OD_{600}$ and pH. Based on the data, growth and pH curves were generated.

**Heat resistance characteristics.** Under aseptic condition, the target bacterial solution after 24 h of culturing was put into four sterilized centrifuge tubes; the tubes were placed at room temperature, and respectively bathing in 80˚C, 90˚C, and 100˚C water for 10 min. Following the exposure to various temperatures, the bacteria were cultured in solid RCM for 48 h and then counted. Taking the number of colonies in the room temperature treatment group as 100%, the ratio of other groups to it is calculated as the survival rate.

**Gastric acid and bile salt tolerance.** The pH of artificial gastric juice was adjusted to 1.5, 2.5, and 3.5 with 5 M HCl [12]. 10 mL fresh culture was centrifuged at 3,000 rpm for 10 min, then the pelleted cells were suspended in 10 mL pH 1.5, 2.5, and 3.5 artificial gastric juice and incubated in 37˚C for 1 h, 2 h, and 3 h respectively. After incubation, samples were washed three times with phosphate-buffered saline (PBS; pH 7.2) before serially diluted up to $10^{-8}$, 50 μL of the diluted cultures was spread on RCM agar plates and incubated for 48 h at 37˚C, 6 replicates for each sample, followed by colony counting. The survival rate (%) in gastric acid was calculated with the mean of colony counts of the plates after treatment with respect to their mean before treatment. The results were presented in mean number of surviving bacteria ± standard deviation [13].

In the bile salt tolerance study, the concentrations of bovine bile salt in liquid medium were 1.0 g/L, 1.5 g/L, 2.0 g/L, 2.5 g/L, 3.0 g/L, 3.5 g/L, and 4.0 g/L. After sterilization at 120˚C for 20 min, the target strain was inoculated according to 1% inoculation volume. Sealing with liquid paraffin, cultured at 37˚C for 24 h before the $OD_{600}$ was measured.

## Safety test

**Animals.** A total of 162 35-day-old Rex rabbits (half males and half females) were randomly divided into 3 groups with 9 replicates per group and 6 rabbits per replicate. Rabbits were housed in cages (1 female and 1 male rabbit/cage) and the trial lasted 35 days. This test was carried out in Weinan New Rex rabbit Farm, Weinan (China).

*Clostridium sartagoforme* was added into drinking water. A, B and C respectively represent the amount of bacteria added according to the body weight of Rex rabbits: $10^8$ CFU/Kg, $10^6$ CFU/Kg and medium mixture. Adding amount was based on the related studies of *Clostridium butyricum* [14].

The diarrhea rate and mortality were tested. Diarrhea rate: the number of diarrhea per day in each treatment group was recorded (diarrhea, too large or too small feces and jelly-like feces were regarded as diarrhea), and the diarrhea rate of the experimental rabbits was calculated by the following calculation method [15]:

$$\text{Diarrhea rate} = \frac{\text{Cumulative days of diarrhea rabbits in each replicate}}{\text{Number of rabbits in each replicate} \times \text{Feeding days}} \times 100\%$$

**Ethic statement and animal experiments permission.** The protocol was approved by the Committee on the Ethics of Laboratory Animals of Northwest Agriculture & Forest University (approved number: 0369/2018).

## Statistical analysis

The experimental data were analysed using SPSS 22.0 software (SPSS Inc., Chicago, IL, USA), and one-way analysis of variance were used to examine the gastric acid tolerance of *Clostridium sartagoforme*, mortality and diarrhea rate of rabbits. The results were expressed as mean ± SEM. A statistical significance was declared at $P < 0.05$ and extremely significant were defined as $P < 0.01$.

## Results

### Isolation and identification of *C. sartagoforme*

According to the morphology results, Gram staining, and microscopic examination of the cultured strains, nine eligible strains were obtained. Using the DNA of each strain as a template, the 16S rDNA was amplified using universal primers, and the amplified products were electrophoresed on a 1% agarose gel. The electrophoresis results of the target strain XN-T4 are shown in Fig 1. PCR for the strains and the amplified products were sequenced to identify their specific species. The results are shown in Table 2.

The target strain XN-T4 16S rDNA was sequenced with a length of 1,432 bp and the accession number is MW450913.

Pairwise sequence alignment of the sequence to BLAST showed that the strain XN-T4 was closely related to *C. sartagoforme* (99% homology over a 100% sequence query cover). Several strains with higher homology were selected to make a phylogenetic tree. The sequences were aligned using MEGA (6). Phylogenetic analysis using maximum likelihood. The results are shown in Fig 2. Therefore, XN-T4 could be considered *C. sartagoforme*.

### Morphological characteristics

*C. sartagoforme XN-T4* was cultured on solid RCM for 24 h. The morphology is shown in Fig 3. The colony was white, round with neat edges, moist, smooth, and opaque, with a diameter of 0.5 mm-2.5 mm. Gram staining was positive, bacteria occurred as single cells or in pairs, the spores were terminal, the bacteria were drumstick-shaped when spores were formed, the diameter of the bacteria was 0.5 μm-1.0 μm × 2.0 μm-6.3 μm, and the spore diameter was 1 μm-1.2 μm × 1 μm-1.2 μm.

### Physiological and biochemical characteristics

*C. sartagoforme XN-T4* could decompose glucose, sucrose, and galactose, but not arabinose. It could not use inositol or dulcitol, it did not liquefy gelatin, and the reaction of lysine decarboxylase was positive. These results are shown in Table 3.

### Fiber hydrolysis characteristics

*C. sartagoforme XN-T4* produced clear and visible hydrolysis circles on CMC-Congo Red plate (Fig 4), ratio of the hydrolytic circle and the strains colony is approximately 2.5. This indicating that the bacteria could decompose and utilize cellulose and secrete cellulase outside of the cell.

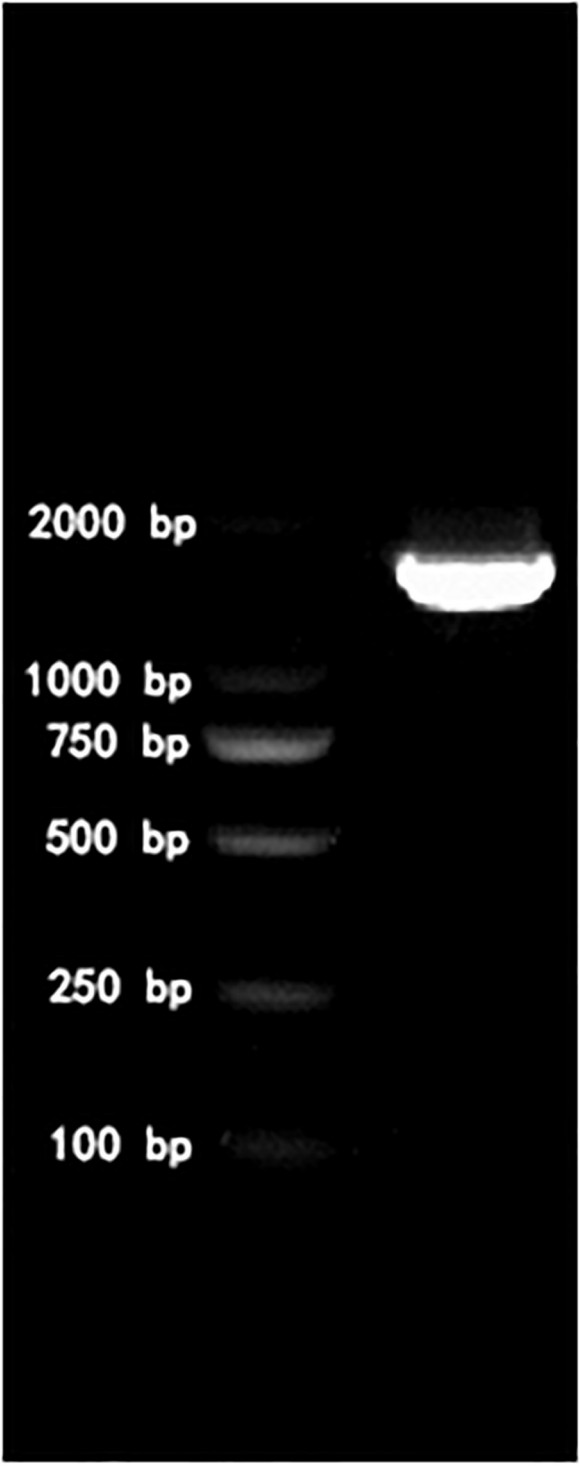

**Fig 1. Agarose gel electrophoresis of *C. sartagoforme XN-T4* 16S rDNA.**

**Table 2. Identification of the isolated bacteria.**

| Strain Number | Strain identified |
|---|---|
| XN-T14 | *Clostridium sordellii* |
| XN-D1 | *Clostridium tertium* |
| XN-D2 | *Clostridium tertium* |
| XN-D3 | *Clostridium tertium* |
| XN-T4 | *Clostridium sartagoforme* |
| XN-D5 | *Clostridium tertium* |
| XN-B2 | *Clostridium sordellii* |
| XN-B3 | *Clostridium tertium* |
| XN-B5 | *Clostridium tertium* |

## Growth characteristics

The growth curve of *C. sartagoforme XN-T4* is shown in Fig 5. The lag phase was about 5 h after inoculation, and then it entered the log phase, which lasted about 5 h before entering the stable phase. The pH also started to drop rapidly at 5 h; this drop lasted for about 10 h after which the pH stabilized at about 4.9.

## Thermal tolerance

*C. Sartagoforme XN-T4* had good heat resistance. The survival rate exceeded 85% after treatment at 80˚C for 10 min, but the survival rate dropped rapidly to less than 10% after exposure to 90˚C. All strains died after treatment at 100˚C (Table 4).

## Gastric acid and bile salt tolerance

The results showed that the survival rate of *C. sartagoforme XN-T4* treated at pH 1.5 for 1 h was more than half, and that for 3 h under pH 2.5 and pH 3.5 was more than 50%, as shown in Table 5.

C. sartagoforme XN-T4 had good bile salt tolerance. It grew well in the liquid medium with 0.15% bile salt content. It still grew in the liquid medium with 0.3% bile salt content, but could not grow in the medium with 0.35% or more bile salt content (Table 6).

## Safety test

There were no significant differences in diarrhea rate and mortality among groups A,B and C, as shown in Table 7.

## Discussion

### Isolation and identification of *C. sartagoforme XN-T4*

In this experiment, *C. sartagoforme* was only isolated from Hyla rabbit, but not from Rex or Hycole rabbits. This is likely due to the species of rabbit, age, and food ingredients in the region [16]. Bogatyrev et al. pointed out that the composition of the host's intestinal flora depends on the host's species, diets, age, and gender [17].

C. sartagoforme can produce spores. Accordingly, the sample was heated and pretreated to kill the non-spore bacteria, and the target strain was isolated by repeated streaking inoculation and purification. According to the colony morphology and Gram staining characteristics, the cultured strains were selected. Combined with sugar and alcohol fermentation experiments,

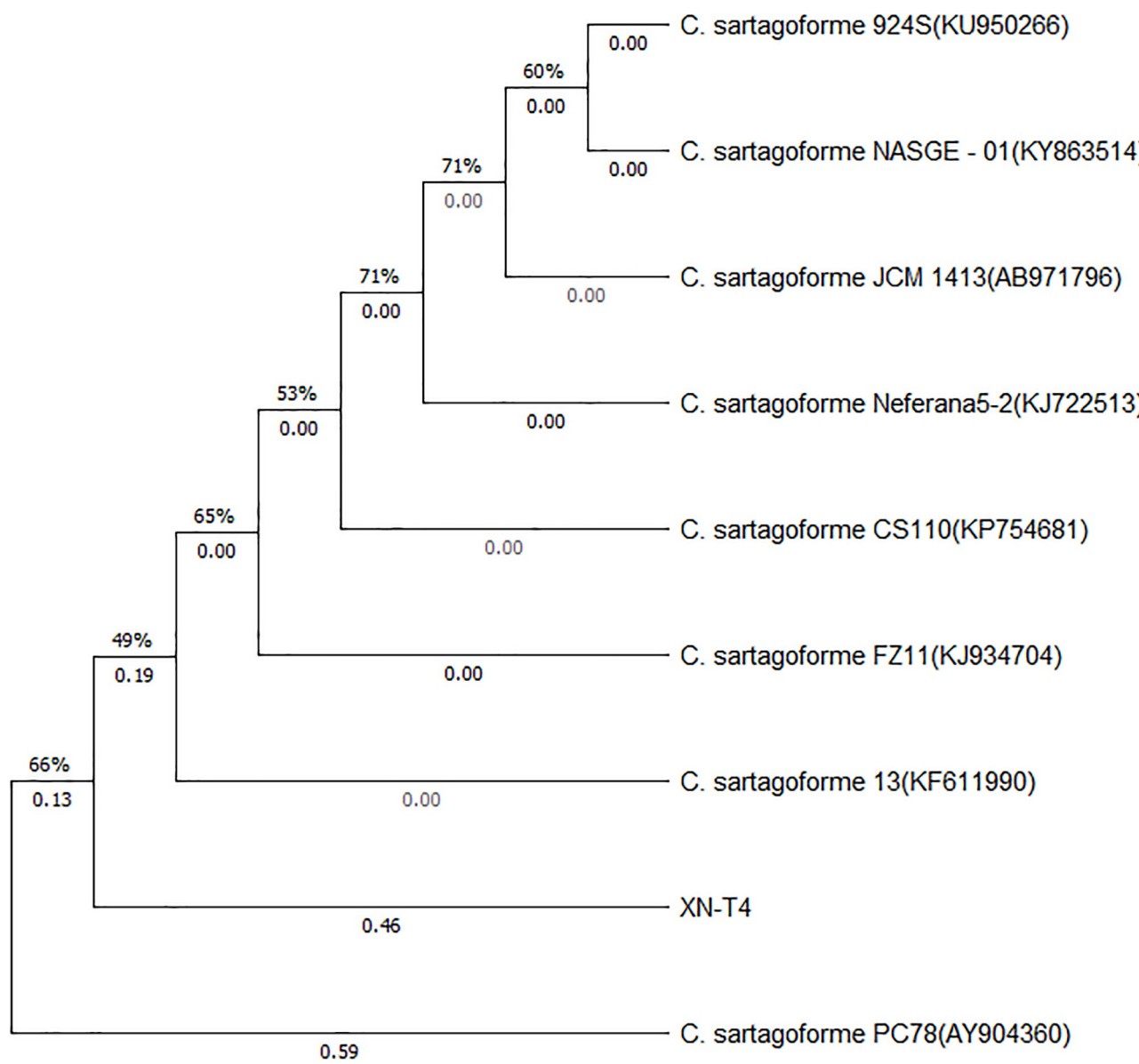

**Fig 2. Phylogenetic tree of *C. sartagoforme XN-T4* 16S rDNA.**

gelatin liquefaction experiments and molecular identification, the strain was identified as *C. sartagoforme* [18].

16S rDNA is the gene of the 16S rRNA subunit of bacterial ribosomes. As the most commonly used molecular identifier in bacterial taxonomy, it is about 1,500 bp. It not only reflects the differences between different bacterial genera, but also facilitates sequence determination [19]. This study directly performed 16S rDNA sequence determination on the selected strains on the basis of the selection of colony morphology and strain Gram staining characteristics. Gelatin liquefaction and sugar fermentation tests were performed as auxiliary identification, which effectively reduced the workload and the test cost, and the target strain was accurately identified.

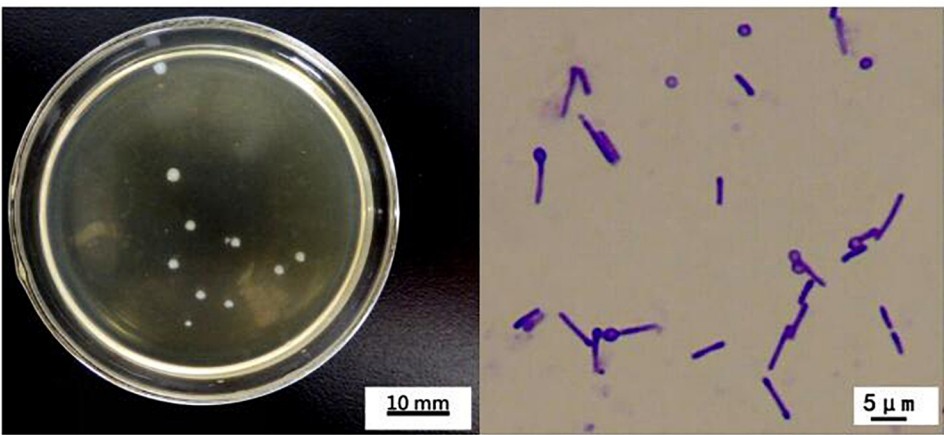

**Fig 3. Colonial and cell morphology of *C. sartagoforme* XN-T4.**

## Physiological and biochemical characteristics of *C. sartagoforme* XN-T4

Bacterial spores have thick spore walls, multiple spore membranes, a solid structure, low water content (80% of propagules and 40% of spores), an almost stopped metabolism, and a strong refractive index [20]. Because the spores produce a unique dipicolinic acid, combined with calcium, the spores have a strong resistance to adverse physical and chemical environmental conditions; they are especially resistant to high temperature, drying and osmotic pressure, and ordinary chemicals do not easily penetrate them [20]. The *C. sartagoforme* XN-T4 isolated in this experiment can form spores to resist adverse environmental conditions, which is beneficial to the production and preservation of the strain.

C. sartagoforme* can use a variety of sugars and alcohols as carbon sources. The strain isolated in this experiment could decompose mannitol, but could not use arabinose or inositol. This is consistent with the results of the API bacterial identification system. The *C. sartagoforme* isolated from the rumen of buffalo is contrary to the results of this experiment [10], which may be due to the fact that many physiological and biochemical characteristics of bacteria are encoded by extrachromosomal genetic factors. The factors that affect the expression of physiological and biochemical characteristics are more complicated [12]; therefore, the results were different. *Clostridium sartagoforme XN-T4* can liquefy gelatin, which is consistent with the previous reports [21, 22].

**Table 3. Identification of the physiological characteristics of *C. sartagoforme* XN-T4.**

| Items | Results | Items | Results |
|-------|---------|-------|---------|
| Arabinose | – | Mannitol | + |
| Mannose | + | Sorbitol | + |
| Glucose | + | Inositol | – |
| Galactose | + | Dulcitol | – |
| Fructose | + | Lysine decarboxylase | + |
| Rhamnose | + | Gelatin hydrolysis | – |
| Sucrose | + | | |

Notes: + indicates that the result is positive, – indicates that the result is negative.

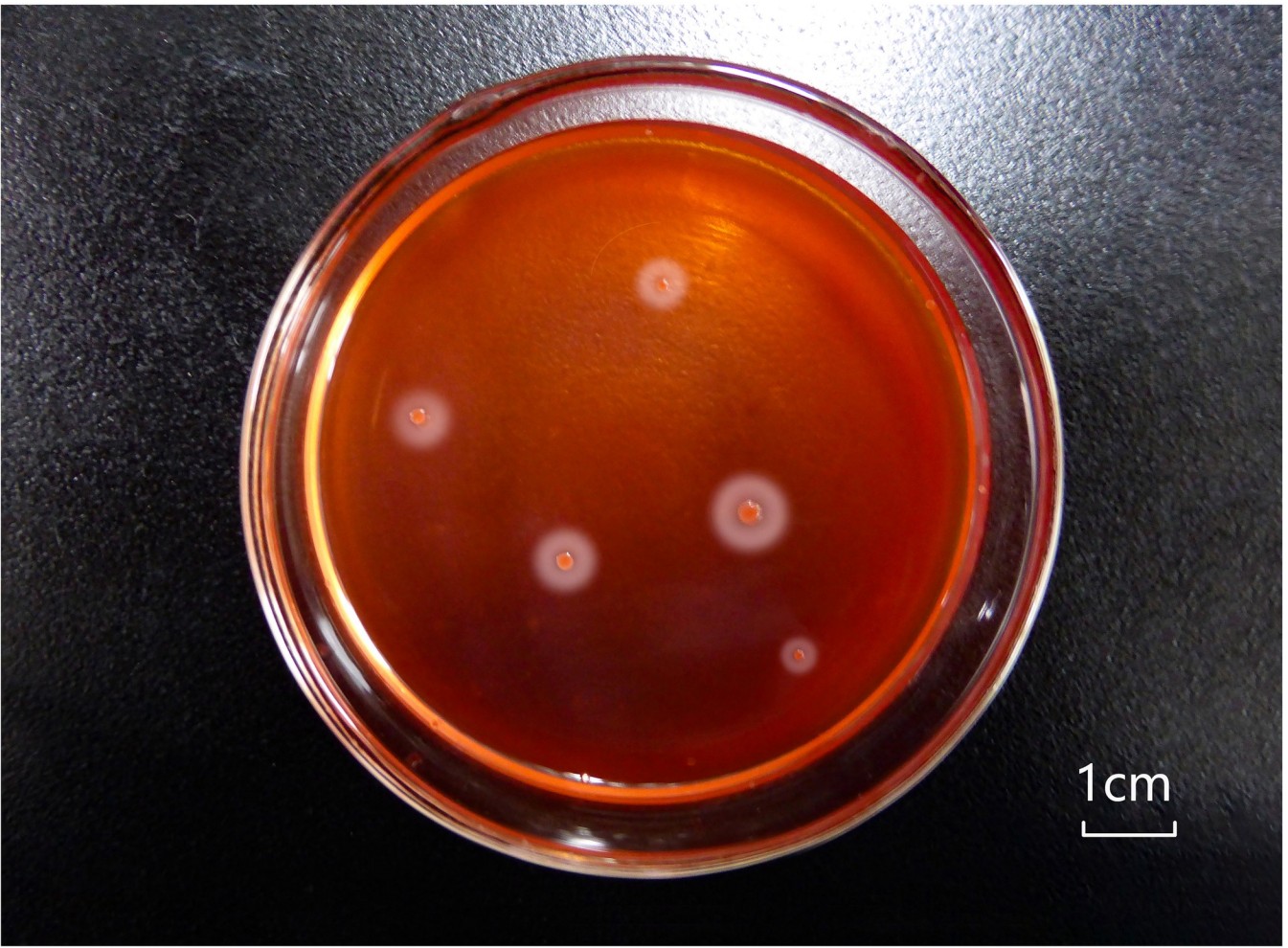

**Fig 4. Characteristic of *C. sartagoforme XN-T4* on Congo-CMC.**

Although rabbits are herbivores, they cannot secrete cellulase themselves, they mainly rely on cecal symbiotic microorganisms to decompose cellulose in the feed [1]. These microorganisms decompose cellulose and produce volatile fatty acids(VFA) for rabbits to use [23]. In addition, due to the coprophagy of rabbits, microorganisms can provide certain bacterial proteins for domestic rabbits [24]. The transparent circle on the CMC-Congo Red plate indicates that the bacterium can decompose cellulose, it means that *C. sartagoforme XN-T4* has the potential to improve the digestibility of cellulose for herbivores as a microbial feed additive.

### Safety, growth characteristics and stress resistance of *C. sartagoforme XN-T4*

There is no report that *C. sartagoforme* is a pathogen, and the data of *C. sartagoforme XN-T4* in animal feeding experiments shows that this strain has no adverse effects on the health of rabbits. This demonstrates that the strain is safe to a certain extent. In addition, the composition of primary metabolites and secondary metabolites should be further studied to confirm its security.

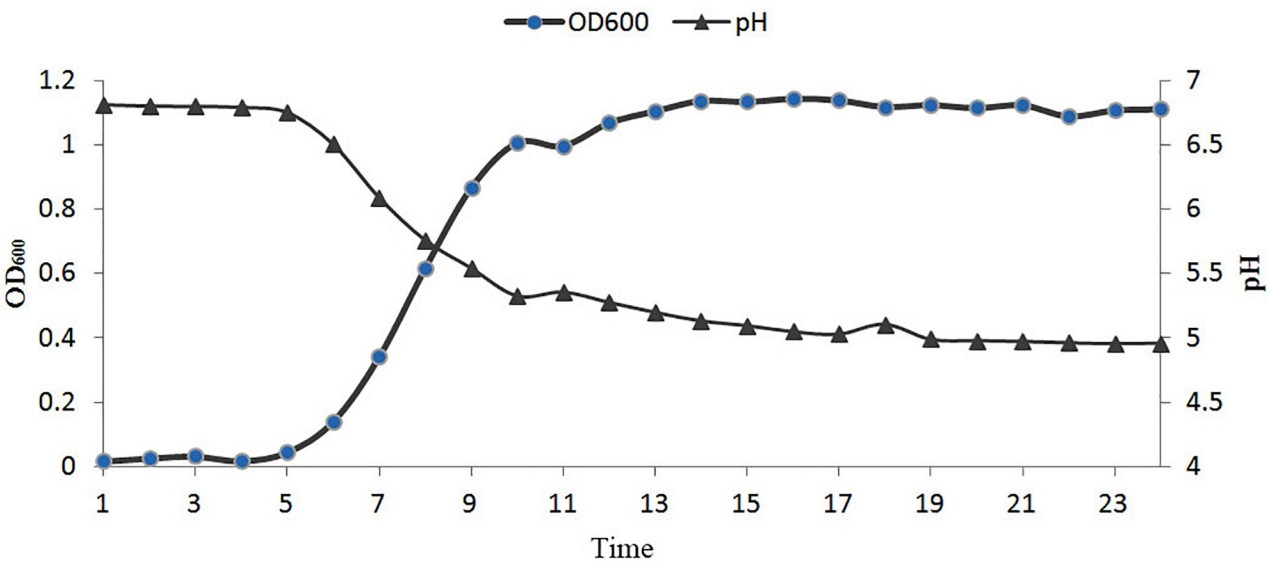

**Fig 5. Growth and pH curve of *C. sartagoforme* XN-T4.**

The growth of bacteria can be divided into the delay phase, logarithmic phase, constant phase, and decay phase [25]. The number of newly propagated cells and the number of dead cells in the constant phase are basically in a stable state, and the total number of viable bacteria is at the highest level. This stage is the best time to obtain bacteria [26]. *C. sartagoforme* XN-T4 began to enter the logarithmic phase about 5 h after inoculation, the logarithmic phase lasted for about 5 h, and then entered the constant phase. Through the growth curve of *C. sartagoforme* XN-T4, it is possible to know which growth period the bacteria are in. In the preparation of microbial additives, the timing of termination of culture can be accurately selected to maximize the yield of bacteria.

In the growth curve of *C. sartagoforme* XN-T4, after entering the logarithmic phase, the pH of the fermentation broth continued to decrease, indicating that volatile fatty acids were

**Table 4. Survival rate of *C. sartagoforme* XN-T4 at different temperatures after 10 min.**

|  | Temperature | | |
|---|---|---|---|
|  | **80˚C** | **90˚C** | **100˚C** |
| **Survival rate (%)** | 85.55±8.82 | 8.06±1.70 | 0 |

**Table 5. Survival rate of *C. sartagoforme* XN-T4 in artificial gastric juice.**

| Conditions | | Hours | | |
|---|---|---|---|---|
|  |  | **1** | **2** | **3** |
| **Survival rate (%)** | pH = 1.5 | 58.65±4.22[a] | 37.54±4.29[b] | 31.09±7.34[b] |
|  | pH = 2.5 | 79.95±2.32[A] | 51.56±2.26[B] | 53.39±3.76[B] |
|  | pH = 3.5 | 88.97±0.83[A] | 74.75±1.39[B] | 58.33±1.54[C] |

Notes: Different lowercase letters on the upper right shoulder of the same row indicate significant differences ($P < 0.05$), and uppercase letters indicate extremely significant differences ($P < 0.01$).

**Table 6. *C. sartagoforme XN-T4* bile salt tolerance.**

| Content of bile salt | 0.10% | 0.15% | 0.2% | 0.25% | 0.3% | 0.35% | 0.4% |
|---|---|---|---|---|---|---|---|
| OD$_{600}$ | 1.01±0.17 | 1.01±0.20 | 0.33±0.16 | 0.27±0.11 | 0.11±0.08 | 0 | 0 |

**Table 7. Effects of *C. sartagoforme XN-T4* on the diarrhea rate and mortality of Rex rabbits (%).**

| | A | B | C |
|---|---|---|---|
| *Diarrhea rate* | 0.69±0.18[a] | 0.85±0.09[a] | 0.74±0.33[a] |
| **Mortality** | 1.85±5.6[a] | 1.85±5.6[a] | 1.85±5.6[a] |

Notes: A, B and C respectively represent the amount of bacteria added according to the body weight of Rex rabbits: 10$^8$ CFU/Kg, 10$^6$ CFU/Kg and medium mixture. Different lowercase letters on the upper right shoulder of the same row indicate significant differences (P < 0.05).

produced during the growth of the bacteria [25]. A study showed that the VFA produced during the growth of *C. sartagoforme* are mainly acetae, formiate, butyrate and lactate [8]. VFA accounts for 40% of rabbit maintenance energy requirement [27].

*C. sartagoforme XN-T4* could tolerate 80˚C for 10 min and ensured that the number of viable bacteria was not less than 80%, indicating that the strain had certain heat resistance. However, as the temperature reached 90˚C, the number of viable bacteria decreased rapidly. Microbial additives can be added by mixing in feed or drinking water. Because rabbits do not like to eat powder or wet feed and have rodent behavior, pellet feed is usually used. The temperature of pellet feed in the circular mould pelleting machine is often above 80˚C. Therefore, if *C. sartagoforme XN-T4* is used as the microbial additive, the method of adding drinking water is more appropriate.

Gastric juice in the stomach and bile salt in the small intestine can kill the vast majority of bacteria in food and protect the health of the body. In resting gastric glands, the intragastric pH is approximately 3–4 [28, 29], and gastric secretion is neither influenced by food intake nor fasting [30]. The strain can not only survive in the medium with bile salt concentration of 0.3%, but also maintain a certain degree of growth, indicating that the bacterium has good tolerance to bile salt, which is similar to the results of Wang TH on *Clostridium butyricum* [15], which belongs to the same genus of *Clostridium*. When microbial additives are used in monogastric animals such as rabbits, it is necessary to ensure that a sufficient number of strains reach the cecum before their beneficial effects can be achieved. *C. sartagoforme XN-T4* has good tolerance to gastric juice and bile salt. The survival rate of *C. sartagoforme XN-T4* treated at pH 3.5 for 3 h was more than half, and it could grow in the medium with bile salt concentration of 0.3%. This indicated that the strain could tolerate the action of gastric juice and bile salt and reach the hindgut.

## Conclusion

A strain of *Clostridium sartagoforme XN-T4* was isolated from Hyla rabbit feces by RCM medium. Through colony characteristics and cell morphology, combined with physiological and biochemical methods and PCR identification. The strain has good stress resistance, can decompose cellulose, and has no obvious toxic effect on rabbits. Theoretically, this bacteria has a strong ability to resist the effects of gastric acid and bile salts in animal digestive juices. It may be a candidate for microbial additives to improve the digestibility of cellulose in rabbits.

Futher more, for its secondary metabolite composition, safety and in vitro and in vivo metabolic characteristics still need experimental research.

## Supporting information

**S1 Data.**
(XLS)

**S1 Fig.**
(TIF)

**S1 Raw images.**
(PDF)

## Acknowledgments

We appreciate the manager of the Rabbit Farm, Liu Xinle, for his support and help with this study.

## Author Contributions

**Conceptualization:** Ruiguang Gong, Xiangyang Ye, Zhanjun Ren.

**Data curation:** Ruiguang Gong, Shuhui Wang.

**Formal analysis:** Ruiguang Gong.

**Funding acquisition:** Zhanjun Ren.

**Investigation:** Ruiguang Gong, Xiangyang Ye.

**Methodology:** Ruiguang Gong, Xiangyang Ye, Shuhui Wang.

**Project administration:** Ruiguang Gong, Zhanjun Ren.

**Software:** Ruiguang Gong, Xiangyang Ye.

**Writing – original draft:** Ruiguang Gong.

**Writing – review & editing:** Ruiguang Gong.

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
