## [Decision Letter · Decision Letter 0]

27 Oct 2020

PONE-D-20-28305

Isolation, identification and biological characteristics of Clostridium sartagoforme from rabbit

PLOS ONE

Dear Dr. Gong,

Thank you for submitting your manuscript to PLOS ONE. After careful consideration, we feel that it has merit but does not fully meet PLOS ONE’s publication criteria as it currently stands. Therefore, we invite you to submit a revised version of the manuscript that addresses the points raised during the review process.

We look forward to receiving your revised manuscript.

Kind regards,

Dr Alistair K Brown

Academic Editor

PLOS ONE

Journal Requirements:

'The funders had no role in study design, data collection and analysis, decision to publish, or preparation of the manuscript.'

*Please include your amended statements within your cover letter; we will change the online submission form on your behalf.*

Reviewers' comments:

Reviewer's Responses to Questions

**Comments to the Author**

1. Is the manuscript technically sound, and do the data support the conclusions?

Reviewer #1: Yes

Reviewer #2: Partly

2. Has the statistical analysis been performed appropriately and rigorously? 

Reviewer #1: Yes

Reviewer #2: Yes

3. Have the authors made all data underlying the findings in their manuscript fully available?

Reviewer #1: Yes

Reviewer #2: Yes

4. Is the manuscript presented in an intelligible fashion and written in standard English?

Reviewer #1: No

Reviewer #2: Yes

5. Review Comments to the Author

Reviewer #1: This manuscript reports the isolation, identification and biological characteristics of Clostridium sartagoforme from rabbit. The study is original. However, it is necessary to add more data to make the study solid from the scientific viewpoints. In particular, the authors should add more research data on the probiotic effect and safety of Clostridium sartagoforme in vitro. In addition, the following points need to be revised and clarified.

1. English usage should be checked thoroughly by native users. There is still room to improve the language and the grammatical errors in the manuscript need to be corrected.

2. In Abstract, “a bacillus-producing strain was isolated from the feces of Hyla rabbit using reinforced clostridium medium (RCM).” Bacillus and Clostridium are two different genera. Confused, please explain it and rewrite this sentence. The abstract section does not render the general significance and conceptual advance of the work clearly accessible to a broad readership, and the logic of the abstract is confused.

3. In Short title, “Characteristics of rabbit Clostridium sartagoforme”, please rewrite.

4. Please insert line numbers to ensure easy referencing during the reviewing process.

5. The Introduction should state the purpose of the investigation and give a short review of the pertinent literature, please include more details.

6. In Isolation and culture methods, “The beaker was placed in a 80 ℃ water bath for 10 minutes to kill the non-spore bacteria;” Is the water bath time too short?

7. Identification of C. Sartagoforme should include morphological identification, physiological and biochemical characterization, and molecular identification. Please clarify this issue and make appropriate adjustments to the content of the section.

8. In Heat resistance characteristics, “The survival rate was calculated as the ratio of other groups to the colony count of 100%. The calculation method of survival rate is unclear, please rewrite.

9. In “Gastric acid and bile salt tolerance”: 1) “and then the samples were inoculated in solid RCM and cultured anaerobically for 48 h before colony count”. This description is unclear. What’s the method? Plate count method? Please describe correctly. 2) The method of bile salt tolerance study is incorrect, please explain. The authors only determined the growth status of C. sartagoforme XN-T4 in different concentrations of bile salt solutions. It is not enough. Please add the survival rate of C. sartagoforme XN-T4 treated with different concentrations of bile salt solution for different time.

10. In Statistical analysis, what is the repeating unit of the experiment? What’s the expression of the results? The results were calculated only by SPSS? Please add the information.

11. In Isolation and identification of C. sartagoforme, the 16S rDNA gene sequence of the strain should be submitted to GenBank to obtain accession number. The 16S rDNA gene sequence and BLAST result of the strain can be deleted. When constructing the phylogenetic tree, what is the method? Please add confidence in the phylogenetic tree and make a clear figure.

12. Fig 5, the authors wrote “Characteristic of C. sartagoforme XN-T4 on Congo-CMC.” Is this 5 duplicates? Why are the sizes of the hydrolysis circles different? Please explain. It was strongly suggested that the authors listed the cellulose hydrolysis zone of C. sartagoforme.

13. In Growth characteristics, “this drop lasted for about 9 h” change to “this drop lasted for about 10 h”.

14. In Discussion, “Based on sugar alcohol fermentation experiments and gelatin liquefaction experiments for molecular identification, the isolated bacteria was further identified as C. sartagoforme (Shen and Chen., 2016).” Confused, please rewrite this sentence.

15. The citation of references is not standardized, please correct it.

16. In Discussion, “bile salt in bile accounts for 1% - 2%”. However, what the authors tested in the experiment was the growth of C. sartagoforme XN-T4 in 0.1% to 0.4% bile salt solution. It is recommended that the authors test the survival rate of C. sartagoforme XN-T4 after being treated in 1-2% bile salt solution for a certain period of time.

Reviewer #2: Thank you for the opportunity to review this manuscript. The authors describe the first, to the best of my knowledge, known isolation of Clostridium sartagoforme from the stool of a Hyla rabbit and suggest its potential utility as a microbial feed additive. Previously, Clostridium sartagoforme has been isolated from the groundwater, human stool samples, soil samples, and buffalo rumen. There is a lot of impressive work within the following document to identify the isolate as Clostridium sartagoforme, yet there is a lack of direction in the manuscript. What was the purpose of targeting Clostridium species in general? Why was this specific colony morphology selected? What motivated the authors to search for a microbial feed additive for rabbits? Additionally, the majority of the discussion section restates results but does little to contextualize this data towards the stated goal of developing a feed additive. While the data is thorough, the lack of explanation regarding the motivation of the study and minimal discussion limits the ability of the manuscript to effectively communicate with the reader.

Reviewer Comments:

1. Introduction

1.1. The initial setup of this paper relating to microbial feed additives was intriguing. However, following this introduction, there is little mention of this same notion. I will address this further down in the manuscript, but there is a lack of a major theme for this manuscript. What were the motivations for isolating Clostridium sartagoforme?

2. Materials and Methods

2.1. Overall, the methods seem appropriate.

2.2. Paragraph 5; line 2 -- It is stated: “the colonies that met the characteristics continued to be anaerobic cultured with RCM…” Could you please expand upon which characteristics you were looking for? I assume you were looking for common characteristics of Clostridium sartagoforme, but this has not been made clear. This may be further helped through further expansion of the introduction section as mentioned above.

2.3. Paragraph 7; line 4 -- For further reproducibility, please consider providing detail into the specific 2X mix used in the amplification of the 16S rRNA gene.

2.4. Table 1 -- Please revise the formatting of the PCR reaction conditions for readability

3. Results

3.1. Could the authors lend some insight into their naming conventions for each strain?

3.2. Table 6: How was the determination made between “++ good growth” and “+ growth”? Can the authors please clarify or provide a quantitative measure for how this determination was made?

4. Discussion

4.1. Paragraph 2; line 6 -- Up until this point, the authors have utilized numbered citations. However, the citation here is formatted differently. The journal requires numbered citations, please revise to match journal requirements.

4.2 Paragraph 7; line 6 – The authors state that the identification of the growth characteristics of C. sartagoforme XN-T4 “provide an important guarantee for the next step of scientific research and reasonable preparation of microbial agents for production tests”. The meaning behind this sentence is ambiguous and difficult to discern. Please consider revising for clarity.

4.3 Overall, there are many valuable threads of knowledge in the discussion section. However, there is a lack of a unifying flow of the data. In its current state, the discussion reads much like a rehashing of the results with the occasional relating to previous work with C. sartagoforme. While the authors state in the abstract that their study provides evidence of the potential utility of C. sartagoforme as a microbial food additive, there is essentially no discussion of the results that help support this point. Additionally, and even more critically, there is little discussion as to why C. sartagoforme would be beneficial as a microbial food additive.

6. PLOS authors have the option to publish the peer review history of their article (what does this mean?). If published, this will include your full peer review and any attached files.

Reviewer #1: No

Reviewer #2: No

---

## [Author Response · Author response to Decision Letter 0]

21 Mar 2021

Response to reviewers-3

1.Ren Zhanjun was the specific recipient of the five fundings [ (1) Shaanxi Province Science and Technology Project (K3310216062); (2) Shaanxi Province Agricultural Science and Technology Innovation and Research Project (16NY-108); (3) Shaanxi Province Agricultural Science and Technology Innovation Transformation Project (NYKJ-2020-YL-16); (4) Yangling Demonstration Zone Industry-University-Research Collaborative Innovation Major Project (1017cxy-15); (5) Key R&D project of Shaanxi Province (2018ZDXM-NY-041)] .

2.The methods of animal experiments were accordance with the relevant regulations by the Committee on Laboratory Animals of Northwest Agriculture & Forest University (NO. 0369/2018), and the pdf of animal experiments permission had been added as “supporting information”.

---

## [Decision Letter · Decision Letter 1]

5 May 2021

PONE-D-20-28305R1

Isolation, identification and biological characteristics of Clostridium sartagoforme from rabbit

PLOS ONE

Dear Dr. Ren,

Thank you for submitting your manuscript to PLOS ONE. After careful consideration, we feel that it has merit but does not fully meet PLOS ONE’s publication criteria as it currently stands. Therefore, we invite you to submit a revised version of the manuscript that addresses the points raised during the review process.

We look forward to receiving your revised manuscript.

Kind regards,

Alistair K Brown

Academic Editor

PLOS ONE

Reviewers' comments:

Reviewer's Responses to Questions

**Comments to the Author**

1. If the authors have adequately addressed your comments raised in a previous round of review and you feel that this manuscript is now acceptable for publication, you may indicate that here to bypass the “Comments to the Author” section, enter your conflict of interest statement in the “Confidential to Editor” section, and submit your "Accept" recommendation.

Reviewer #1: All comments have been addressed

Reviewer #2: (No Response)

2. Is the manuscript technically sound, and do the data support the conclusions?

Reviewer #1: No

Reviewer #2: Partly

3. Has the statistical analysis been performed appropriately and rigorously? 

Reviewer #1: Yes

Reviewer #2: No

4. Have the authors made all data underlying the findings in their manuscript fully available?

Reviewer #1: Yes

Reviewer #2: Yes

5. Is the manuscript presented in an intelligible fashion and written in standard English?

Reviewer #1: Yes

Reviewer #2: No

6. Review Comments to the Author

Reviewer #1: (No Response)

Reviewer #2: Thank you for the opportunity to once again review this manuscript. The authors have made a lot of improvements with this revision. The addition of the rabbit experiments is a fantastic addition to this manuscript and helps to move this manuscript beyond a potential use in a live animal model and probes at whether it is safe in trials involving rabbits. Despite the improvements, this manuscript is not yet ready for publication. One of the most glaring omissions is in the references provided. There are numerous instances throughout the manuscript that would benefit from the addition of some further literature review. Additionally, some further explanation and clarification are required before this manuscript will be ready for evaluation.

Abstract: It would benefit the paper if information concerning the rabbit testing of C. sartagoforme was added to the abstract

Line 50: Please consider adding a reference for this sentence.

Line 51: Please consider adding a reference for this statement.

Line 54: Please consider adding a reference for this statement concerning Bacillus.

Line 107: Please consider adding a reference for this statement. It may also be prudent to further explain this and the mechanism (over-decolorization) in some more detail.

Line 112: For the sake of reproducibility, could the authors please provide some further details on the genomic DNA extraction kit that was used?

Line 162: This difficult is difficult to read and should be rewritten (especially the end) for further clarity.

Line 172: It is unclear to me exactly how many rabbits were in each experimental group. Please consider revising this section for more clarity. A simple figure may also help this portion be more easily interpreted.

Line 193: Can you please describe the statistical tests that were used to compare the control and treatment groups for the trials including the rabbits?

Table 3: C. sartagoforme should have returned a negative result for the utilization of sorbitol. Were any steps taken to control for potential contamination in the experiments?

Line 282: As mentioned previously, how were significant differences between treatment groups of rabbits evaluated?

Table 7: It is curious that the diarrhea rate and mortality rate for group A are identical. Additionally, the standard error of the means appears to be very large for the diarrhea rate, this causes some concern for the conclusions that are drawn here. This concern may be alleviated once the analysis performed is described in more detail.

Line 294: It would be beneficial to add a citation for this sentence.

Line 303: The use of the term “molecular clock” is interesting here. I’m not sure that this is the best term to describe this. It seems that “molecular barcode” or “molecular identifier” would be some potential alternatives.

Line 314: It would be beneficial for a citation to be added here.

Lines 330-334: The manuscript would be improved if some references were added for each of these statements.

Line 334: There appears to be some typos and confusion in this sentence.

Line 341: It would be beneficial for a citation to be added here.

Line 346: It would be beneficial for a citation to be added here.

Line 355: It would be beneficial for a citation to be added here.

Line 371-374: Both of these references are for studies involving mice. Do these statements hold true for rabbits as well?

Line 385: This concluding sentence seems weirdly informal and would benefit from some rewriting.

7. PLOS authors have the option to publish the peer review history of their article (what does this mean?). If published, this will include your full peer review and any attached files.

Reviewer #1: No

Reviewer #2: No

---

## [Author Response · Author response to Decision Letter 1]

16 Jul 2021

Response to reviewers-4

Abstract: It would benefit the paper if information concerning the rabbit testing of C. sartagoforme was added to the abstract

The sentence of “Moreover, animal testing indicated that this strain has no adverse effects on the morbidity and mortality of rabbits.” has been added.

Line 50: Please consider adding a reference for this sentence.

Citations has been added:

[1] Ayyat MS, Al-Sagheer AA, Abd El-Latif KM, Khalil BA. Organic Selenium, Probiotics, and Prebiotics Effects on Growth, Blood Biochemistry, and Carcass Traits of Growing Rabbits During Summer and Winter Seasons. Biol Trace Elem Res. 2018 Nov;186(1):162-173. doi: 10.1007/s12011-018-1293-2. Epub 2018 Mar 7. PMID: 29516355.

[2]Pogány Simonová M, Chrastinová Ľ, Lauková A. Autochtonous Strain Enterococcus faecium EF2019(CCM7420), Its Bacteriocin and Their Beneficial Effects in Broiler Rabbits-A Review. Animals (Basel). 2020 Jul 14;10(7):1188. doi: 10.3390/ani10071188. PMID: 32674281; PMCID: PMC7401553.

[3]Elghandour MMY, Tan ZL, Abu Hafsa SH, Adegbeye MJ, Greiner R, Ugbogu EA, Cedillo Monroy J, Salem AZM. Saccharomyces cerevisiae as a probiotic feed additive to non and pseudo-ruminant feeding: a review. J Appl Microbiol. 2020 Mar;128(3):658-674. doi: 10.1111/jam.14416. Epub 2019 Sep 8. PMID: 31429174.

[4]Zhou Y, Ni X, Wen B, Duan L, Sun H, Yang M, Zou F, Lin Y, Liu Q, Zeng Y, Fu X, Pan K, Jing B, Wang P, Zeng D. Appropriate dose of Lactobacillus buchneri supplement improves intestinal microbiota and prevents diarrhoea in weaning Rex rabbits. Benef Microbes. 2018 Apr 25;9(3):401-416. doi: 10.3920/BM2017.0055. Epub 2018 Jan 30. PMID: 29380642.

[5]Wang J, Ni X, Wen B, Zhou Y, Liu L, Zeng Y, Zhao W, Khalique A, Wang P, Pan K, Yu Z, Jing B, Liu H, Zeng D. Bacillus strains improve growth performance via enhancing digestive function and anti-disease ability in young and weaning rex rabbits. Appl Microbiol Biotechnol. 2020 May;104(10):4493-4504. doi: 10.1007/s00253-020-10536-9. Epub 2020 Mar 19. PMID: 32193576.

[6]El-Deep MH, Dawood MAO, Assar MH, Ahamad Paray B. Aspergillus awamori positively impacts the growth performance, nutrient digestibility, antioxidative activity and immune responses of growing rabbits. Vet Med Sci. 2021 Jan;7(1):226-235. doi: 10.1002/vms3.345. Epub 2020 Sep 9. PMID: 32902158; PMCID: PMC7840208.

Line 51: Please consider adding a reference for this statement.

Citations has been added:

[7]Wang Z. Preparation of Lactic acid bacteria Preparation an its Preliminary Application in Weaned Piglets [dissertation]. Harbin: Northeast Agricultural University; 2019.

[8]Bintsis T. Lactic acid bacteria as starter cultures: An update in their metabolism and genetics. AIMS Microbiol. 2018 Dec 11;4(4):665-684. doi: 10.3934/microbiol.2018.4.665. PMID: 31294241; PMCID: PMC6613329.

Line 54: Please consider adding a reference for this statement concerning Bacillus.

Citation has been added:

[9]Hu JH. Handbook of Medication. 4th ed. China Beijing: Jundun Press; 2009. Chinese.

Line 107: Please consider adding a reference for this statement. It may also be prudent to further explain this and the mechanism (over-decolorization) in some more detail.

Citation has been added:

[12]Shen P, Chen XD. Microbiology. 8th ed. Beijing: Higher Education Press; 2016. Chinese.

Explain: Clostridium is generally positive for Gram staining, but the cell wall structure of the old bacteria would become loose if the culture time is too long, and the result of Gram negative may appear after staining.

Line 112: For the sake of reproducibility, could the authors please provide some further details on the genomic DNA extraction kit that was used?

Details on the bacteria genomic DNA extraction kit that was added：Takara Bio Inc., Shiga, Japan.

Line 162: This difficult is difficult to read and should be rewritten (especially the end) for further clarity.

The methods were clarified as：

After incubation, bacteria were washed three times with phosphate-buffered saline (PBS; pH 7.2), 10-fold serial dilutions were made, and then, colony-forming units were counted on RCM agar plates after 48 h of incubation at 37 ℃. Tolerance was expressed as a percentage of the number of colony-forming units after incubation in artificial gastric acid relative to that of the normal RCM.

Line 172: It is unclear to me exactly how many rabbits were in each experimental group. Please consider revising this section for more clarity. A simple figure may also help this portion be more easily interpreted.

The methods were clarified as：

The experiment was conducted in a completely randomized design with 3 treatments, and every treatment with 3 replicates, 18 rabbits in each replicate (162 Rex rabbits 35-day-old ) that included the following treatments.

Line 193: Can you please describe the statistical tests that were used to compare the control and treatment groups for the trials including the rabbits?

Mortality and diarrhea rate of rabbits were analysised by one-way analysis of variance with SPSS 22.0 software (SPSS Inc., Chicago, IL, USA).

Table 3: C. sartagoforme should have returned a negative result for the utilization of sorbitol. Were any steps taken to control for potential contamination in the experiments?

The bacteria were inoculated in the clean bench, and the biochemical reaction tube was liquid sealed with sterilized liquid paraffin and covered with parafilm.

Line 282: As mentioned previously, how were significant differences between treatment groups of rabbits evaluated?

Data of mortality and diarrhea rate have been reanalyzed, and the significant differences between treatment groups of rabbits have been marked.

Table 7: It is curious that the diarrhea rate and mortality rate for group A are identical. Additionally, the standard error of the means appears to be very large for the diarrhea rate, this causes some concern for the conclusions that are drawn here. This concern may be alleviated once the analysis performed is described in more detail.

Data of mortality and diarrhea rate have been reanalyzed, but the standard error of the means still large, especially the numerical value of mortality are the same in different treament, the reason is that only 1 rabbit died in each group. Raw Data of Mortality and Diarrhea Rate as follow:

Raw Data of Mortality and Diarrhea Rate

Total number of rabbits Group Mortality Diarrhea rate

162 A Replicate 1 5.56 0.79

 Replicate 2 0 0.79

 Replicate 3 0 0.48

 B Replicate 1 5.56 0.95

 Replicate 2 0 0.79

 Replicate 3 0 0.79

 C Replicate 1 0 1.11

 Replicate 2 5.56 0.63

 Replicate 3 0 0.49

Notes: A, B and C respectively represent the amount of bacteria added according to the body weight of Rex rabbits: 108 CFU/Kg, 106 CFU/Kg and medium mixture. 

Line 294: It would be beneficial to add a citation for this sentence.

Citation has been added:

[18]Sun XM. Effect of L-arginine and N-carbamylgluatmate on small intestinal villi development and cecal microbiota composition of young rabbits [dissertation]. Changchun: JiLin University; 2019.

Line 303: The use of the term “molecular clock” is interesting here. I’m not sure that this is the best term to describe this. It seems that “molecular barcode” or “molecular identifier” would be some potential alternatives.

Molecular clock had been changed as molecular identifier.

Line 314: It would be beneficial for a citation to be added here.

Citation has been added:

[20]Chen JD, Huang QY. Animal husbandry microbiology. 6th ed. China Beijing: Agriculture Press; 2017. Chinese.

Lines 330-334: The manuscript would be improved if some references were added for each of these statements.

Citations has been added:

Line 330:

[24]Ouyang WQ. Animal Physiology. Beijing: Science Press; 2021. Chinese.

Line 331:

[25]Guo ZQ, Wang B, Kuang LD, Yang R, Li CY, Zheng J. Gastrointestinal development and cecal cellulase activity in Qixing and New Zealand rabbits. Chinese Journal of Animal Science. 2021,57(03):130-133. Chinese

Line 332:

[26]Simonová MP, Chrastinová L, Kandricáková A, Kubašová I, Formelová Z, Chrenková M, Miltko R, Belzecki G, Strompfová V, Lauková A. Enterocin M and Sage Supplementation in Post-weaning Rabbits: Effects on Growth Performance, Caecal Microbiota, Fermentation and Enzymatic Activity. Probiotics Antimicrob Proteins. 2020 Jun;12(2):732-739. doi: 10.1007/s12602-019-09584-z. PMID: 31414382.

Line 334:

[27]Gu ZL, Qing YH, Ren KL. China rabbit Science. China Beijing: China Agriculture Press; 2013. Chinese.

Line 334: There appears to be some typos and confusion in this sentence.

This sentence has been changed as:

The transparent circle on the CMC-Congo Red plate indicates that the bacterium can decompose cellulose, it means that C. sartagoforme XN-T4 has the potential to improve the digestibility of cellulose for herbivores as a microbial feed additive. 

Line 341: It would be beneficial for a citation to be added here.

Citation has been added:

[28]Dong XZ, Cai MY. System Identification Manual of Common Bacteria. Beijing: Science Press; 2001. Chinese.

Line 346: It would be beneficial for a citation to be added here.

Citation has been added:

[29]Yu LJ. Principles and Technology of Fermentation Engineering. Beijing: Higher Education Press; 2021. Chinese.

Line 355: It would be beneficial for a citation to be added here.

Citation has been added:

[29]Yu LJ. Principles and Technology of Fermentation Engineering. Beijing: Higher Education Press; 2021. Chinese.

Line 371-374: Both of these references are for studies involving mice. Do these statements hold true for rabbits as well?

These studies are mainly on mice, and there are no rabbits, so I have to refer to mice here.

Line 385: This concluding sentence seems weirdly informal and would benefit from some rewriting.

The conclusion rewrited as:

A strain of Clostridium sartagoforme XN-T4 was isolated from Hyla rabbit feces by RCM medium. Through colony characteristics and cell morphology, combined with physiological and biochemical methods and PCR identification. The strain has good stress resistance, can decompose cellulose, and has no obvious toxic effect on rabbits. It may be a candidate for microbial additives to improve the digestibility of cellulose in rabbits. Futher more, for its secondary metabolite composition, safety and in vitro and in vivo metabolic characteristics still need experimental research.

Reviewers' comments:

1.This manuscript reports the isolation, identification and biological characteristics of Clostridium sartagoforme from rabbit. However, it is not rigorous and unscientific. For using as microbial feed additive, the authors should add more research data on the probiotic effect and safety of Clostridium sartagoforme in vitro.

Dear reviewer, I would like to further the research on this strain. However, as I have graduated, I have no conditions and equipment to invest in this field. At present, I am applying for a doctor's degree, after getting the offer, I plan to continue the research on the probiotic effect and safety of Clostridium sartagoforme in vitro.

2. In Introduction, the authors propose that Clostridium sartagoforme had potential properties as a microbial food additives. However, following this introduction, there is little mention of this same notion. The authors should give a review of the pertinent literature, please include more details. What were the motivations and purpose for isolating Clostridium sartagoforme?

The motivations and purpose were to isolate a strain suitable for rabbits, which can resist the effect of digestive juice and improve the digestibility of cellulose in rabbits. By adding this strain, the economic benefit of rabbit breeding can be improved.

3. In Discussion, while the authors state in the abstract that their study provides evidence of the potential utility of Clostridium sartagoforme as a microbial food additive, there is essentially no discussion of the results that help support this point. Additionally, and even more critically, there is little discussion as to why Clostridium sartagoforme would be beneficial as a microbial food additive.

The reason for Clostridium sartagoforme would be beneficial as a microbial food additive as fellow: 

The strain has no obvious toxic effect on rabbits, to a certain extent, it indicates that the bacterium is not a pathogen.

The strain has good stress resistance which can resist the effect of digestive juice, this means that the bacteria may reach the rabbit's cecum to play a role.

The strain can decompose cellulose, it is possible that the strain can improve feed utilization by decomposing cellulose.

4. All taxa names (species names, genus names, and names of higher categories) should be italicized.

Taxa names had been italicized in the article.

---

## [Decision Letter · Decision Letter 2]

23 Aug 2021

PONE-D-20-28305R2

Isolation, identification and biological characteristics of Clostridium sartagoforme from rabbit

PLOS ONE

Dear Dr. Ren,

Thank you for submitting your manuscript to PLOS ONE. After careful consideration, we feel that it has merit but does not fully meet PLOS ONE’s publication criteria as it currently stands. Therefore, we invite you to submit a revised version of the manuscript that addresses the points raised during the review process.

We look forward to receiving your revised manuscript.

Kind regards,

Alistair K Brown

Academic Editor

PLOS ONE

Journal Requirements:

Reviewers' comments:

Reviewer's Responses to Questions

**Comments to the Author**

1. If the authors have adequately addressed your comments raised in a previous round of review and you feel that this manuscript is now acceptable for publication, you may indicate that here to bypass the “Comments to the Author” section, enter your conflict of interest statement in the “Confidential to Editor” section, and submit your "Accept" recommendation.

Reviewer #1: All comments have been addressed

Reviewer #2: All comments have been addressed

2. Is the manuscript technically sound, and do the data support the conclusions?

Reviewer #1: Partly

Reviewer #2: Yes

3. Has the statistical analysis been performed appropriately and rigorously? 

Reviewer #1: Yes

Reviewer #2: Yes

4. Have the authors made all data underlying the findings in their manuscript fully available?

Reviewer #1: Yes

Reviewer #2: Yes

5. Is the manuscript presented in an intelligible fashion and written in standard English?

Reviewer #1: Yes

Reviewer #2: Yes

6. Review Comments to the Author

Reviewer #1: (No Response)

Reviewer #2: (No Response)

7. PLOS authors have the option to publish the peer review history of their article (what does this mean?). If published, this will include your full peer review and any attached files.

Reviewer #1: No

Reviewer #2: No

---

## [Author Response · Author response to Decision Letter 2]

7 Oct 2021

Response to Reviewer#5:

Point 1: In the last review, I commented “the authors should add more research data on the probiotic effect and safety of Clostridium sartagoforme in vitro”. The author did not realize the importance of its safety and effectiveness. In any case, as microbial a feed additive, the safety of probiotics is very important. If there is no guarantee that Clostridium sartagoforme is safe, all subsequent studies will be futile. Therefore, some basic safety tests such as antibiotic susceptibility test, hemolytic test, and toxicity test are necessary.

Response 1: Thank you for your instructive suggestions. The studies of diarrhea rate and mortality rate on rabbits were used to demonstrate the safety of this strain, and further experiments such as cellulase production, components of secondary metabolites, antibiotic susceptibility test, hemolytic test, toxicity test, blood biochemical inedxes and organ coefficients of experimental animals will be carry out.

Point 2: In the last review, I commented “The authors should give a review of the pertinent literature, please include more details.” in the introduction. The author did not make changes in accordance with the comments, and the explanation given was unreasonable. The introduction should provide with sufficient background information to introduce the reader into the reported study. If the author is not clear about the relevant research progress of Clostridium sartagoforme, how does the author think that it can be used as a microbial food additive? A review of the pertinent literature is necessary and it can well reflect the authors' motivations and purpose for isolating Clostridium sartagoforme.

Response 2: Thank you for your careful reading of our manuscript. The introduction has been rewrote as follow:

Rabbits are herbivores, there are abundant cellulose degrading bacteria resources in rabbit intestinal tract, because they rely on the microorganisms in the caecum to digest cellulose but not produce cellulase by themselves. In addition, cellulose degrading bacteria isolated from animals have natural advantages in the utilization of crude fiber feed. Most of them are inherent bacteria in the intestinal tract, which are easier to adapt to the intestinal environment of the animal and play a role.

Clostridium sartagoforme is an anaerobic spore producing bacterium, which determines that the bacterium has a natural advantage in resisting the effects of digestive juice and adapting to the anaerobic environment of the digestive tract. Researches have shown that the strain can decompose microcrystalline cellulose, carboxymethyl cellulose and chitin, and its metabolites include formic acid, acetic acid, propionic acid, lactic acid, butyric acid, isovaleric acid and hydrogen. Simůnek J et al. isolated C. sartagoforme from feces of horse, and this strain has high exocellular activity of N-Acetylglucosaminidase. Zhang JN isolated Clostridium sartagoforme FZ11 from cow dung compost, using urea as a nitrogen source and raw corn stalk as a substrate, confirming that this strain could decompose cellulose and have good hydrogen production characteristics. In Nathani Neelam M and coleague’s research, Clostridium sartagoforme AAU1 was isolated from Surti buffalo rumen liquor, and they pointed out that VFAs produced by the bacteria play an important role in animal health maintenance through genomic analysis. At present, the research about C. sartagoforme isolated from digestive tract or feces of rabbits has not been reported, and there is little information on the biological characteristics of this strain. In this experiment, C. sartagoforme was isolated and identified from rabbit feces, and its physiological, biochemical and stress resistance characteristics were preliminarily studied, which help to provide reference for the further development of rabbit microbial additives.

Point 3-(1): Whether there are other more reasonable explanations for changing from Gram-positive to Gram-negative of Clostridium sartagoforme.

Response 3-(1): Thank you for your careful reading of our manuscript. For this phenomenon, only this point of view is currently being seen.

Point 3-(2): In Line 162, reviewers-4 commented “This difficult is difficult to read and should be rewritten (especially the end) for further clarity”, the authors clarified the methods, but there are still some details that you need to add. After 10-fold serial dilutions, how many dilutions should be taken, whether it is spread or spotted, and how is the colony-forming units calculated?

Response 3-(2): I'm sorry for the article writing problem. Samples were diluted up to 10-8, 50 μL of the diluted cultures was spread on RCM agar plates and incubated for 48 h at 37 ℃, 6 replicates for each sample, followed by colony counting. The survival rate (%) in gastric acid was calculated with the mean of colony counts of the plates after treatment with respect to their mean before treatment. The results were presented in mean number of surviving bacteria ± standard deviation.

Point 3-(3): In Line 172, reviewers-4 commented “It is unclear to me exactly how many rabbits were in each experimental group.”, but there are still some details that you need to add. Are female rabbits or male rabbits used in the experiment, and what is the ratio?

Response 3-(3): I apologize for the ambiguity in the description. A total of 162 35-day-old Rex rabbits (half males and half females) were randomly divided into 3 groups with 9 replicates per group and 6 rabbits per replicate.

Point 3-(4): The results of diarrhea rate and mortality are quite different before and after, please confirm the accuracy and authenticity of the data. The formula for calculating the diarrhea rate may be wrong, please confirm.

Response 3-(4): Thank you for your valuable advice. The diarrhea rate has been checked to be correct, but the formula for diarrhea rate is indeed not accurate enough, it has been changed as:

Point 3-(5): In Line 371-374, reviewers-4 commented “Both of these references are for studies involving mice. Do these statements hold true for rabbits as well?”, the authors responded “These studies are mainly on mice”. Please change references for studies related to rabbits.

Response 3-(5): Thank you for your careful work. The reference about “gastric juice and pH in stomach” had been changed to those researches on rabbits. The results of this study are indeed very different in rabbits and mice. I am ashamed of the previous erroneous quotation of references and sincerely thank you for your suggestions. 

Point 3-(6): Reviewers-4 suggested that the author added a lot of references, but most of the newly added references are in Chinese. Please replace some of the references with English references.

Response 3-(6): According to the reviewer’s comments, most of the references in Chinese had been changed to English references.

---

## [Decision Letter · Decision Letter 3]

26 Oct 2021

Isolation, identification and biological characteristics of Clostridium sartagoforme from rabbit

PONE-D-20-28305R3

Dear Dr. Ren,

We’re pleased to inform you that your manuscript has been judged scientifically suitable for publication and will be formally accepted for publication once it meets all outstanding technical requirements.

Kind regards,

Alistair K Brown

Academic Editor

PLOS ONE

Additional Editor Comments (optional):

Reviewers' comments:

Reviewer's Responses to Questions

**Comments to the Author**

1. If the authors have adequately addressed your comments raised in a previous round of review and you feel that this manuscript is now acceptable for publication, you may indicate that here to bypass the “Comments to the Author” section, enter your conflict of interest statement in the “Confidential to Editor” section, and submit your "Accept" recommendation.

Reviewer #1: All comments have been addressed

2. Is the manuscript technically sound, and do the data support the conclusions?

Reviewer #1: Yes

3. Has the statistical analysis been performed appropriately and rigorously? 

Reviewer #1: Yes

4. Have the authors made all data underlying the findings in their manuscript fully available?

Reviewer #1: Yes

5. Is the manuscript presented in an intelligible fashion and written in standard English?

Reviewer #1: Yes

6. Review Comments to the Author

Reviewer #1: (No Response)

7. PLOS authors have the option to publish the peer review history of their article (what does this mean?). If published, this will include your full peer review and any attached files.

Reviewer #1: No

---

## [Editor Report · Acceptance letter]

5 Nov 2021

PONE-D-20-28305R3 

Isolation, identification, and biological characteristics of *Clostridium sartagoforme* from rabbit 

Dear Dr. Ren:

I'm pleased to inform you that your manuscript has been deemed suitable for publication in PLOS ONE. Congratulations! Your manuscript is now with our production department. 

Kind regards, 

on behalf of

Dr. Alistair K Brown 

Academic Editor

PLOS ONE